

# Vibration error compensation algorithm in the development of the laser interference absolute gravimeter

**Qiong Wu[1], Yuntian Teng[1], Xiaomei Wang[1], Yanxiong Wu[1,2], and Yang Zhang[1]**

[1]Institute of Geophysics, China Earthquake Administration, Beijing, 100081, China

[2]Institute of Disaster Prevention, Sanhe, 065201, China

*Correspondence to:* Qiong Wu ( wuqiong@cea-igp.ac.cn)

**Abstract.** Measurement error arising from vibration interference is recognized as the primary obstacle limiting the accuracy and stability of laser interference absolute gravimeters. The present work addresses this issue by proposing a global search optimization algorithm that determines the optimal

absolute value of gravity based on the measured time-displacement coordinates of a falling body and the signal obtained from the passive vibration isolation system of the inertial reference corner-cube in a laser interference absolute gravimeter. Results of numerical calculations conducted under vibration interference conditions with added white noise resulting in a signal-to-noise ratio of 40 dB demonstrate the following. (1) The accuracy and standard deviation of the gravimeter obtained using the proposed

algorithm are −0.04 μGal (1 μGal = 1×10$^{-8}$m/s2) and 0.24 μGal, respectively, while those values obtained by the standard least-squares solution are 10.19 μGal and 154.11 μGal, respectively. (2) The resolution of the test results shows that the average response of the reference value of acceleration due to gravity superimposed by a disturbance of 1.00 μGal is 1.01 μGal using the proposed algorithm and 0.87 μGal using the standard least-squares solution.

**1 Introduction**

Laser interference absolute gravimeters are high-precision gravimetry instruments that are widely used in various fields such as metrology (Ouedraogo., et al 2007), solid geophysics (D'Agostino., et al, 2008), seismic observation (Timofeev., et al, 2018; Zhang., et al, 2019), and geodesy (Xing., et al, 2017). These instruments measure the vertical position of a free-falling test body in a drop chamber under vacuum

using a laser interferometer that generates an optical interference fringe each time the test body falls a distance equal to one-half of the wavelength of the laser light, and the timing of the fringes is determined



from an atomic clock, which thereby provides highly precise time-displacement coordinates to determine the acceleration of the test body due to gravity.

However, vibrations caused by the servo-motor driven mechanism in laser interference gravimeters and

environmental vibrations are coupled during the measurement process and introduce vibration error into the measurement result by affecting the inertial state of the reference corner-cube (Wu., et al, 2012). This vibration error has become the primary obstacle limiting the accuracy and stability of laser interference absolute gravimeters, and affects absolute gravity measurement instruments based on the principle of atomic interference (Wu., et al, 2019; Ménoret., et al, 2018).

Numerous approaches have been developed for addressing the critical issue of vibration error. For example, the Micro-g LaCoste FG5 absolute gravimeter based on laser interferometry has obtained nearly ideal vibration error handling using a delicate design mechanism denoted as a superspring (Rinker., 1983; Hinderer., et al, 2002). However, this solution suffers from a complicated structure and inflexibility for extending its application range. Other approaches developed for this purpose mainly include

decreasing the self-system vibration of the instrument (Niebauer., et al, 2011), vibration isolation technology (Nelson., et al, 1991; Newell., et al, 1997; Richman., et al, 1998; Li., et al, 2014), and vibration error compensation algorithms (Long., et al, 2012; Wang., et al, 2017). One of the simplest and most feasible technologies employs a passive vibration isolation system (PVIS) to suspend the reference corner-cube (Zumberge., et al, 1986). However, current practical applications of the PVIS remain far

from meeting the requirements of instrument stability, accuracy, and precision under different observation conditions.

This issue is addressed in the present work by applying the PVIS in conjunction with a specifically developed vibration error compensation algorithm. To this end, we first analyze the PVIS of the reference corner-cube to obtain the transfer function of the system as a vibration isolation device and a signal

detection device, and thereby establish a vibration interference error model. Then, the error model is employed to develop a new vibration error compensation algorithm (VECA) that conducts a global search based on a genetic algorithm (Wu., et al, 2018) to determine the optimal absolute value of gravity based on the measured time-displacement coordinates of the falling body and the signal obtained from the PVIS of the reference corner-cube in the laser interference absolute gravimeter. Finally, the accuracy, precision,

and resolution of the gravimeter obtained using the proposed algorithm are evaluated by numerical calculations conducted under vibration interference conditions, and the results are compared with those



obtained using the standard least-squares solution (LSS). The results demonstrate that the proposed

algorithm provides a substantial anti-vibration capability and is worthy of applying within absolute

gravimeter designs based on laser interferometry.

## 2 Construction of the vibration error model

### 2.1 System analysis of the PVIS

The main component of a laser interference absolute gravimeter is a drop chamber under vacuum, inside

which the test mass serving as a falling body (FB) is positioned within a cart mechanism that serves to

raise, release, and catch the FB. There is no mechanical connection between the FB and the cart. The cart

mechanism is driven by a servo motor installed on the outside of the drop chamber. After raising the FB

to its initial position, the cart is accelerated downward by the servo motor at an acceleration rate slightly

greater than that due to gravity until the FB is freely falling. Then, the motion of the cart is matched with

the free-falling motion of the FB, and the time-displacement coordinates of the FB relative to the

reference corner-cube are measured. However, the random vibrations arising from the servo motor in the

instrument and human activities within the measurement environment will be introduced during the

measurement process, which therefore couples vibration error with the calculation results for the absolute

value of the acceleration due to gravity.

The PVIS used to reduce the influence of vibration interference on the inertial state of the reference

corner-cube is illustrated schematically in Fig. 1, in which the reference corner-cube and the mass block

of the PVIS assembly of mass M is connected with the ground by a spring of stiffness K and a damping

mechanism with a damping factor σ. Damping is produced by a mechanism employing a coil and

magnetic steel structure that is commonly used in seismometers, where the value of σ is adjusted by

varying the coil current. The displacement x(t) of the assembly relative to the ground is directly output

by a precisely designed differential capacitance sensor and its corresponding circuitry. In addition, the

absolute displacement of the ground is defined as y(t), and the absolute displacement of the assembly

with respect to the earth center of mass is defined as z(t). Finally, we note that the PVIS applies no closed-

loop feedback at any point.



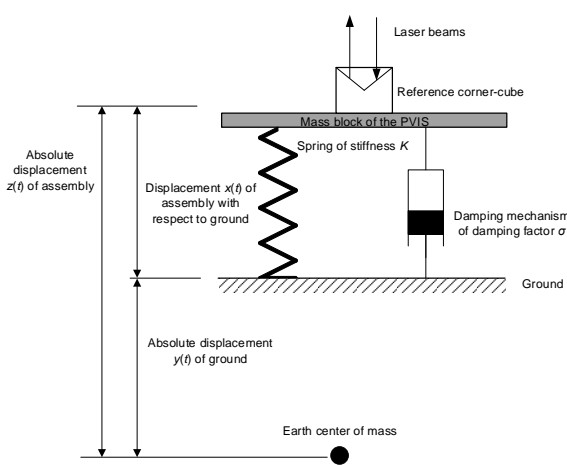

**Figure: 1 Schematic illustrating the passive vibration isolation system (PVIS)**

For conducting system analysis of the PVIS, we first note from Fig. 1 that $x(t) = z(t) - y(t)$ and

arbitrarily define the positive direction as vertically upward from the center of mass of the earth.

Accordingly, the dynamic equation of motion of the mass block of the PVIS and reference corner-cube

assembly is given as follows (Thomson., et al, 1972):

$$M\ddot{z}(t) + \sigma\dot{z}(t) + Kx(t) = 0,\tag{2.1}$$

and its Laplace transform is

$$Ms^2Z(s) + \sigma sZ(s) + KX(s) = 0,\tag{2.2}$$

$$X(s) = Z(s) - Y(s).\tag{2.3}$$

The transfer function of the vibration isolation effect of the PVIS can be given as follows:

$$\Phi_1(s) = \frac{Z(s)}{Y(s)} = \frac{K}{Ms^2+\sigma s+K} = \frac{\omega_0{}^2}{s^2+2\beta s+\omega_0{}^2},\tag{2.4}$$

where $2\beta = \frac{\sigma}{M}$ and the squared intrinsic frequency $\omega_0{}^2 = \frac{K}{M}$. The transfer function of the detection

effect of the PVIS can be given as follows:

$$\Phi_2(s) = \frac{X(s)}{Y(s)} = -1 \cdot \frac{Ms^2+\sigma s}{Ms^2+\sigma s+K} = -1 \cdot \frac{s^2+2\beta s}{s^2+2\beta s+\omega_0{}^2}.\tag{2.5}$$

Applying (2.5) in conjunction with (2.3) yields the following:

$$Z(s) = \left(1 + \frac{1}{\Phi_2(s)}\right)X(s) = -\frac{\omega_0{}^2}{s^2+2\beta s}X(s).\tag{2.6}$$

The attenuation amplitude versus frequency and phase-frequency characteristic curves of $\Phi_1(s)$ and

$\Phi_2(s)$ are plotted in Fig. 2 for K = 50 N/mm, σ = 60, and M = 5 kg. It can be seen from the amplitude-





frequency curve that the vibration isolation operation of the PVIS (i.e., Φ1(s)) attenuates signals in $y(t)$ above the intrinsic frequency. The vibration isolation effect will become increasingly obvious as the frequency of the signals increases above the intrinsic frequency , but these high frequency signals after

attenuation will still be coupled in the measurement system of the absolute gravimeter, and signals below the intrinsic frequency will be directly coupled into the measurement system without attenuation. In terms of the signal detection operation of the PVIS (i.e., Φ2(s)), signals above the intrinsic frequency will be detected without any attenuation, but signals below the intrinsic frequency are attenuated.

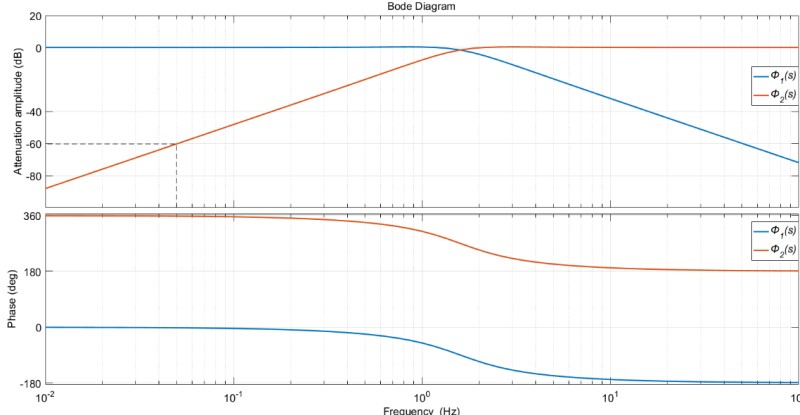

**Figure: 2 Amplitude-frequency and phase-frequency characteristic curves of the vibration isolation operation of the PVIS and its signal detection operation. Here, the phase information is not unwrapped.**

Because of the height restrictions of the drop chamber, the free-falling distance of the FB is controlled within 20 cm, and the free-falling period is about 0.2 s, which can be regarded as the maximum period of the vibration interference signal with a corresponding frequency of 5 Hz. If we consider that signals

with amplitude attenuations greater than −60 dB are effectively attenuated, signals from approximately 0.05 Hz to the intrinsic frequency of the system will still exist in the output signal of the PVIS, although they will be attenuated. Meanwhile, signals below 0.05 Hz can be considered to vary linearly with respect to time during the measurement period of 0.2 s. Additionally, it can be seen from the phase-frequency curves in Fig. 2 that the phases of Φ1(s) and Φ2(s) maintain a consistent relationship. Therefore, the

PVIS can be applied as both a vibration isolation device for the reference corner-cube and a detection sensor of the vibration signal of the reference corner-cube simultaneously. Most of the useful signals related to the disturbance error of the absolute gravity measurement in Z(s) can be recovered by the synchronously output vibration signal X(s).





### 2.2 Mathematical model of the vibration error

In absolute gravimetry, the general method for treating a vertical gradient of the earth's surface is that

the vertical gravity gradient of the measurement point is first assumed to be 0, and then the absolute

acceleration of the effective height of the instrument (Zumberge., 1981; Niebauer., et al, 1989; Timmen.,

2003) is obtained using the absolute gravity measurement instrument. In addition, at least two sets of

relative gravimeters are used to measure the vertical gravity gradient of the measurement point either

before or after conducting the absolute gravity measurement. Finally, the absolute gravity measurement

value of the effective height position is calculated relative to the ground or other height position based

on the obtained vertical gradient. Because this paper mainly analyzes the impact of vibration errors on

gravity measurement results, we adopt the following simplified model for the factors affecting the

measured displacement $S(t_i)$ of the FB as a function of time $t_i$, where $i = 1, 2, \ldots, N$ indexes the time

coordinates established according to the rubidium atomic frequency standard:

$$S(t_i) = S_0 + V_0 t_i + 0.5 g_0 t_i{}^2 + d(t_i) + e(t_i). \tag{2.7}$$

Here, $S_0$ and $V_0$ are the initial displacement and velocity of the FB, $g_0$ is the absolute value of gravity

to be determined from the measurement point, $d(t_i)$ is the vibrational interference coupled to the

measurement system, and e is additive white noise conforming to a normal distribution with a mean of 0

and a variance of 1. Here, $d(t_i)$ is the signal of interest that must be uncoupled from the measurement.

For this purpose, we express the signal $x(t)$ output from the PVIS as

$$x(t_i) = \sum_{k=1}^{M} A_k \cos(\omega_k t_i + \varphi_k) + \sum_{j=0}^{2} p_j t_i{}^j, \tag{2.8}$$

where $A_k$, $\omega_k$, and $\varphi_k$ are the amplitude, angular frequency, and initial phase of the kth harmonic

signal, respectively. In addition, the second term on the right is the trend component with $p_j$ as the

coefficient, where $t_i{}^j$ is $t_i{}^0 = 1$, $t_i{}^1 = t_i$, and $t_i{}^2$, while $j = 0, 1, 2$.

Extracting $d(t_i)$ from $x(t_i)$ requires that we consider the following two important aspects.

①The voltage signal $w(t)$ derived from the PVIS has not been converted into a displacement via

sensitivity calibration. However, the calibration error is on the order of 10-7 m generally, and the error

in the displacement measurement of a microgravity absolute gravimeter must be less than 10-11 m at

least (Christian., 2004). Accordingly, the sensitivity calibration error is much greater than the allowable

error range of a microgravity absolute gravimeter.





② As described in Section 2.1, the vibration signal $x(t)$ is not the absolute displacement of the reference corner-cube $z(t)$, but is rather the result of the system transfer function, as described in equation (2.6), with which the low frequency signals are subject to attenuation.

These issues are addressed to extract $d(t_i)$ from $x(t)$ by introducing comprehensive coefficients $q_{ak}, q_{\varphi k},$ and $q_{pj}$ influencing $A_k, \omega_k,$ and $p_j$ to obtain the following (Akaike., 1980):

$$d(t_i) = \sum_{k=1}^{M} q_{ak} A_k \cos(\omega_k t_i + \varphi_k + q_{\varphi k}) + \sum_{j=0}^{2} q_{pj} p_j t_i^{\,j}. \tag{2.9}$$

This equation can be rewritten as

$$d(t_i) = \sum_{k=1}^{M} (T_{1k}(t_i) R_{1k} + T_{2k}(t_i) R_{2k}) + \sum_{j=0}^{2} q_{pj} p_j t_i^{\,j}, \tag{2.10}$$

where $T_{1k}(t_i) = A_k \cos(\omega_k t_i + \varphi_k),$ $T_{2k}(t_i) = A_k \sin(\omega_k t_i + \varphi_k),$ and the correction coefficients for the harmonic components are $R_{1k} = q_{ak} cos q_{\varphi k}$ and $R_{2k} = -1 * q_{ak} sin q_{\varphi k}$. Next, we apply equation (2.9) to (2.7) as follows:

$$S(t_i) = S_0 + V_0 t_i + 0.5 g_0 t_i^2 + \sum_{k=1}^{M} (T_{1k}(t_i) R_{1k} + T_{2k}(t_i) R_{2k}) + \sum_{j=0}^{2} q_{pj} p_j t_i^{\,j} + e(t_i), \tag{2.11}$$

$$S(t_i) = (S_0 + q_{p0} p_0) + (V_0 + q_{p1} p_1) t_i + (0.5 g_0 + q_{p2} p_2) t_i^2 + \sum_{k=1}^{M} (T_{1k}(t_i) R_{1k} + T_{2k}(t_i) R_{2k}) +$$

$$e(t_i), \tag{2.12}$$

$$S(t_i) = Q_0 + Q_1 t_i + Q_2 t_i^2 + \hat{d}(t_i) + e(t_i), \tag{2.13}$$

where $Q_1 = V_0 + q_{p1} p_1,$ $Q_2 = g_0 + q_{p2} p_2,$ and

$$\hat{d}(t_i) = \sum_{k=1}^{M} T_{1k}(t_i)) R_{1k} + T_{2k}(t_i)) R_{2k}. \tag{2.14}$$

In this case, the term $Q_2$ in (2.13) includes not only the quadratic term arising from acceleration due to

gravity but also the quadratic term in the vibration interference. Therefore, obtaining accurate measurement values for g0 requires that $q_{p2} p_2$ be accounted for as follows:

$$g_0 = 2(Q_2 - q_{p2} p_2). \tag{2.15}$$

**2.3 Analysis of the vibration error compensation**

Comparing (2.11) and (2.13) indicates that, if we can obtain the exact values of $S_0,$ $V_0,$ and $g$ from

(2.13) by some algorithm like a global optimization algorithm, the theoretical parabolic equation can be obtained as

$$\hat{S}(t_i) = S_0 + V_0 t_i + 0.5 g_0 t_i^2, \tag{2.16}$$

and the residual $D_0(t_i)$ can be expressed as

$$D_0(t_i) = S(t_i) - \hat{S}(t_i) = d(t_i) - e(t_i), \tag{2.17}$$



In contrast, if relationship (2.17) is satisfied, the values of $S_0$, $V_0$, and $g$ can be obtained without any

vibration errors. In addition, the satisfaction of (2.17) can be considered as a correlation coefficient of

$D_0(t_i)$ and $d(t_i)$ obtaining a maximum value after de-averaging as follows:

$$\rho_H = \frac{\sum_{k=1}^{n} D_0(t_i) d(t_i)}{\sqrt{\sum_{k=1}^{n} (D_0(t_i))^2 \sum_{k=1}^{n} (d(t_i))^2}}. \tag{2.18}$$

### 3 Proposed VECA and simulation tests

**3.1 Algorithm design**

The solution of the VECA is conducted based on a genetic algorithm. The computational flow of the

VECA for the ith iteration is illustrated in Fig. 3. The relevant parameters employed in the simulation

process are set as follows.

①Interference fringe signal simulation: the fringe signal $U(t_i)$ can be simulated according to the

mathematical form of a fringe signal given by Murata (Murata., 1978) after removing the DC component

and linear drift as follows:

$$U(t_i) = U_0 cos\left(\frac{4\pi}{\lambda} S(t_i)\right), \tag{3.1}$$

Here, $U_0$ is the amplitude of the simulated fringe signal, set as $U_0 = 1$ representative of a normalized

signal, $\lambda$ is the laser wavelength, set as 632.990439 nm, and $S(t_i)$ is the displacement of the FB

defined by (2.7), where $t_i$ is the time of the ith sampling point obtained under an equal sample spacing

at a sampling frequency of 100 MHz. Therefore, the sampling interval is $t_{i+1} - t_i = 1 \times 10^{-8}$ s.

In addition, the absolute value of the acceleration due to gravity $g_0$ is assumed to be 980,110,343.0

µGal in the simulation of the VECA. This term is used to generate the fringe signal required by the

simulation of the VECA and is also used as a reference value to determine the accuracy of the results of

the VECA and LSS calculations.

The anti-interference ability of the VECA is verified by replacing $d(t_i)$ in (2.7) with the two following

vibration interference models in turn:

$$V_1(t) = \sum_{k=1}^{15} A_k \cos(2\pi f_k t), \quad f_k \in (0.05,100] \text{ Hz}, \tag{3.2}$$

$$V_2(t) = \frac{1}{2} p t^2 + q t + b + \sum_{k=1}^{15} A_k \cos(2\pi f_k t), \quad f_k \in (0.05,100] \text{ Hz}, \tag{3.3}$$

where $p = -100 \times 10^{-8} \text{ m/s}^2$, $q = 7 \times 10^{-8} \text{ m/s}$, and $b = 1 \times 10^{-4} \text{ m}$. The frequency range

considered was based on the effective frequency range for actual signal detection in the PVIS and the





characteristic frequency range of natural system vibrations generated during instrument operation. From this range, 15 values of $f_k$ are randomly selected for each calculation, and the corresponding values of $A_k$ are selected randomly as well under the constraint that the amplitude of the interference vibration

signal after component superposition is $3 \times 10\text{-}9$ m. The signal-to-noise ratio of $d(t_i)$ after superimposing $e(t_i)$ is 40 dB.

②Time-displacement coordinates: the time-displacement coordinates are computed by the instantaneous phase of the fringe signal $U(t)$ based on the Hilbert transform with phase unwrapping (Svitlov., et al, 2014; Zumberge., et al, 2004]:

$$S(t) = \frac{\lambda}{4\pi} tan^{-1}\left(\frac{\mathcal{H}\{U(t)\}}{U(t)}\right). \tag{3.4}$$

③Initialize the ergodic range of $\boldsymbol{g}$: first, an initial value of $Q_2$ is obtained by LSS with $S(t)$ defined by (3.4). It is known that the vibration error range introduced by $q_{p2}p_2$ will not exceed $\pm 1000$ μGal under the action of the PVIS in actual measurements [6]. Therefore, we ensure the rationality of the calculation results by setting the ergodic range of $g$ as $Q_2 \pm 5000$.

④Set the parameters of the genetic algorithm: the maximum number of iterations is set as 150, the number of variables is 1, the number of individuals is 80, the number of binary coding digits of the variable is 50, and the generation gap is 0.9.

⑤Design the objective function: the objective function is based on the correlation coefficients between $D_0(t_i)$ and $d(t_i)$ given by (2.18), which is employed to complete the reinsertion, population restoration,

and fitness allocation of the genetic algorithm.

**Figure. 3 Computational flow of the proposed VECA at the ith iteration**





### 3.2 Accuracy and standard deviation of the VECA results

The accuracy of the VECA is calculated as

$$\Delta g = \frac{\sum_{i=1}^{n} g_i}{n} - g_0, \tag{3.5}$$

where $g_i$ is the ith measurement result, and $n$ is the total number of measurements. The standard deviation of the VECA is calculated as

$$\sigma = \sqrt{\frac{\sum_{i=1}^{n}(g_i - g_0)^2}{n-1}}. \tag{3.6}$$

The vibration error suppression effect of the proposed VECA is illustrated for different types of vibration errors based on vibration interference signals (3.2) and (3.3), and the VECA and LSS are applied to each vibration signal 500 times. The values of gi − g0 obtained for each simulation by the VECA and LSS are presented in Fig. 4(a) and (b) for the interference signals given by (3.2) and (3.3), respectively. We note that the VECA results are distributed in the range of ±0.8 µGal, the accuracy is −0.04 µGal, and the

standard deviation is 0.24 µGal. The results of the LSS lie in the range of ±500 µGal, the accuracy is 10.19 µGal, and the precision is 154.11 µGal. Obviously, the proposed VECA improves the accuracy and precision of the measurement results under the condition of vibration interference much better than the LSS.

### 3.3 Resolution testing

The resolution of the algorithm was tested by first calculating the theoretical values of the earth tide $\Delta g_i$ at each minute $i = 1, 2, \dots, 1440$ in Beijing on May 1, 2020 (Munk., et al, 1966). Then, $\Delta g_i$ is added to $g_0$ as

$$g_0' = g_0 + \Delta g_i, \tag{3.7}$$

to approximate the effect of $\Delta g_i$ on the absolute value of acceleration due to gravity in the regional

gravity field at the same measurement period. The values of gi − g0 obtained at each minute by the VECA and LSS are presented in Fig. 5(a) for the interference signals defined in equation (3.3). The resolution of the test results is shown in Fig. 5(b) based on a plotting of the results of $g_i - g_0$ in Fig. 5(a) along the ordinate axis and a plotting of $g_0' - g_0$ along the abscissa axis, and then applying linear regression analysis to the results to obtain the slopes and intercepts of the fitted lines. The system deviation of VECA

is −0.1067 µGal, and the average response of the reference value of acceleration due to gravity

superimposed by a disturbance of 1 µGal is 1.0085 µGal. In contrast, the system deviation of LSS is

−99.0424 µGal, and the average response of the reference value of acceleration due to gravity is 0.8736

µGal, which therefore fails to reflect changes in the acceleration due to gravity under the influence of the

earth tide. Accordingly, we can conclude that the proposed VECA significantly improves the resolution

of the measurement results relative to the LSS.

## 4 Conclusion

Using the PVIS to suspend the reference corner-cube in a laser interference absolute gravimeter is a

simple and feasible technology for reducing random vibration error in gravimetry measurements. A

system analysis of the PVIS demonstrated the feasibility of applying the PVIS for both vibration isolation

and signal detection simultaneously to improve the measurement accuracy of the absolute gravimeter.

Next, a reference corner-cube vibration interference error model was developed and applied to construct

the newly proposed VECA for improving the accuracy, precision, and resolution of the measured value

of gravity using a genetic algorithm. The results of numerical calculations conducted under vibration

interference conditions with added white noise resulting in a signal to noise ratio of 40 dB demonstrated

the following. (1) The accuracy and standard deviation of the gravimeter obtained using the proposed

algorithm are −0.04 µGal and 0.24 µGal, respectively, while those values obtained by the standard LSS

are 10.19 µGal and 154.11 µGal, respectively. (2) The resolution of the test results shows that the average

response of the reference value of acceleration due to gravity superimposed by a disturbance of 1 µGal

is 1.01 µGal using the proposed algorithm and 0.87 µGal using the standard least-squares solution. These

results verify that making full use of the output signal of the PVIS can effectively improve the

performance of the absolute gravimeter via the application of anti-vibration using the VECA. The VECA

has a practical significance in decreasing requirements for vibration isolation hardware and thus

improving the adaptability of the measurement environment in the development of laser interference

absolute gravimeters.





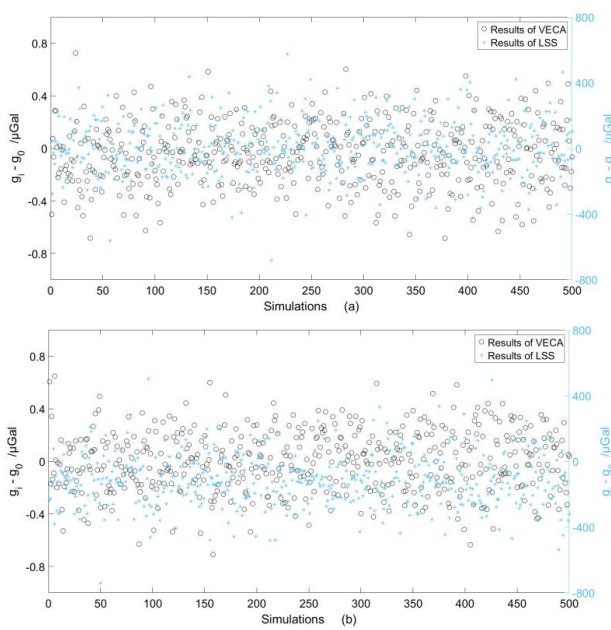


**Figure. 4 Values of gi − g0 obtained by the VECA and LSS for 500 simulations with vibration interference model (3.2) (a) and model (3.3) (b).**

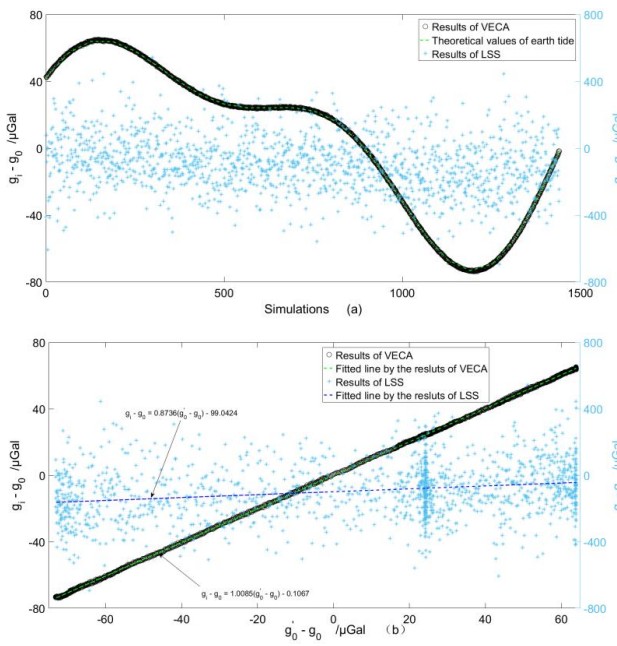

**Figure. 5 Values of gi − g0 obtained for each minute $i = 1, 2, …, 1440$ by the VECA and LSS under the effect of the earth tide in Beijing on May 1, 2020 (a) and a plot of the gi − g0 values in (a) versus the g'i − g0 values along with linear fitting to the results (b).**






**Acknowledgments**

This work is supported by the Special Fund of the Institute of Geophysics, China Earthquake Administration (Grant Number DQJB19B23) and the National Key Research and Development Program of China (2018YFC1503801), the Special Fund of Fundamental Scientific.

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
