# Peer review of "Vibration error compensation algorithm in the development of laser interference absolute gravimeters"

_Geoscientific Instrumentation, Methods and Data Systems, 2020_

## Referee Comment (RC1) · Anonymous Referee #1 · 14 Jan 2021

**General Comments:**

The authors are addressing the problem of the vibration interferences occurring in the laser interference absolute gravimeters. Such kind of gravimeters is using a falling body (FB) to determine the gravity value. The authors suggest to improvements through the use of a passive vibration isolation system (PVIS) combined to a vibration error compensation algorithm (VECA). Indeed, the PVIS has a double usage, reducing vibrations and record the residual vibrations. Then, this residual vibration signal subtracted from the displacement signal of the FB, thanks to the VECA. The authors compared their approach with a standard least squares approach, on 2 different study cases. 1. simu-
lation results by the injection of 2 different noise signals. 2. ability to capture the Earth tide.

The paper addresses a relevant scientific question with new ideas. The use of English is good, and the title reflects the content of the paper. The abstract provides an efficient summary. The approach presented is interesting. However the work and quality are contrasted. Indeed, stronger and more adapted comparisons, would better highlight the work presented. The overall quality of the article good even if the balance between the PVIS and VECA could be improved and some typing errors are present.

**Remarks:**

Since this work is theoretical, the Fig 1. would deserve to include, the laser, the FB, the cart, servomotor, PVIS, reference corner-cube etc, to give a better and faster idea of the suggested set up. The construction of the article is globally correct but could be improved. In section 2.1 the authors are detailing a classical spring-mass-damper, in order to say that some frequencies will be attenuated and a signal recorded. It can be shortened, in my opinion ; especially by comparison with the section 3.1 that would deserve more details. There is a lack of consistency in the mathematical symbols used (multiplication, for example) and citing style. Fig 4 & 5 are not clear, the double scale of the vertical axis makes the reading difficult. It would be clearer with a log scale or zoom box.

**Questions: (specific comments)**

The proposed method is a combination of 2 sub-techniques (PVIS and VECA). But the comparison is global, so it is hard to conclude about the correctness of the sub-technique themselves. What is the sensibility and precision of a PVIS ? Can it be compared alone with other techniques ? What if we add prior form of the noise from PVIS into the LSS ? Can the VECA be compared with other algorithms taking into account the vibrational signal from the PVIS ? I 136. Why add a white noise ? Why with this specific variance ? I 186. Why use the genetic algorithm among many others
? Gradient ? Random walk ? Or Machine Learning ? I 192. Why no noise from the fringe signal is considered ? I 220. What are the input and output of the genetic algorithm ? What is the variable ? Individuals, etc. I 225. What is exactly the objective function chosen which "is based on the correlation coefficients" ? I 226. In figure 3, in "Is i greater than the maximum number of iterations" is the same i-th time iteration or the i-th iteration of the genetic algorithm ? It is confusing. I 236. The LSS used is Linear or Non-linear ?

Technical corrections/typing errors:

- I 102.  $Phi_1$  the subscript missing.
- I 104, unwanted space before the coma.
- I 107. \Phi\_2 the subscript missing.
- l 119. idem
- l 143. k th
- I 147 148. 10 the exponent missing.
- I 161. parentheses for cos and sin functions and multiplication sign
- I 162. it is equation 2.10 instead of 2.9
- I 168. Problems with ))
- I 171. g\_0 the subscript missing.
- I 179. it should d+e instead of d-e?
- I 203 204. [ instead of (
- l 210. power
- I 214. ) instead of ]
I 218. citing style inconsistent

I 237. g\_0 the subscript missing.

l 250. idem
Interactive

comment

---

## Referee Comment (RC2) · Anonymous Referee #2 · 4 Feb 2021

The authors address the issue of vibration errors in absolute gravimeters. They present a vibration error compensation algorithm related to a passive vibration isolation system. I recommend a publication in the Journal Geoscientific Instrumentation, provided the authors address the following points:

(16-17-18) sentence is not clear : the resolution of the tests results shows...the word 'resolution' should be precised as the measuring system resolution. As said, it can be confused with the resolution of the system of equations.

(44) (53-54) (238-240) precise the definition of accuracy and precision to avoid misunderstanding. (66-67) I don't see the relation between the FB –cart motion matching **Printer-friendly version**
and the time-displacement coordinates measurements. A global schematic of the system could help.

(68-71) sentence comprehensible but should be split in two more concise sentences.

(80) applies no closed-loop feedback at any point : need some explanations

(120-121) explain why : Most of the useful signals related to the disturbance error of the absolute gravity measurement in Z(s) can be recovered by the synchronously output vibration signal X(s). in relation with the previous paragraph.

(217) [6] has to be explicited

(219-221) explain the choice of the parameters and the effects in the genetic algorithm.

(222-224) again, a short explanation of genetic algorithm could help the comprehension and shows the advantages of the algorithm with respect to the classical least-squares method.

(269-273) are copy-paste of (13-18), should be reformulated to avoid the exact repetition.

The following are purely typing corrections :

(14) ( 1  $\mu$ Gal = 1×10-8m/s2) the number 2 should be exponent.

(42) technology

(64) servo motor installed on the outside of the drop chamber.

(66-67) will be are introduced.

(74) is are connected

- (93) separate the comma from the equation to avoid confusion with prime.
- (103) no space after 'the intrinsic frequency'

GID
(142) kth harmonic : th should be exponent

(147) 10-7 m : -7 should be exponent

(148) 10-11 m : -11 should be exponent

(153) with which the low frequency signals -> whose low frequency signals

(186) ith iteration : th should be exponent

(194) ith : th should be exponent

(202) and (203) (0.05,100] ->[0.05,100]

(209) 10-9 : -9 should be exponent

(213) ] should be a parenthesis

(226) legend fig 3 : ith iteration, th should be exponent

(231) ith : th should be exponent

GID

---

## Author Comment (AC1) · 15 Feb 2021

Dear Anonymous Referee,

We thank you for your helpful comments and analysis. We have read your suggestions very carefully and have responded to each one below in bold font. All revised portions of the manuscript have been given in red font.

**1, Remarks:**

Since this work is theoretical, the Fig 1. would deserve to include, the laser, the FB, the cart, servomotor, PVIS, reference corner-cube etc., to give a better and faster idea of the suggested set up.

The construction of the article is globally correct but could be improved. In section 2.1 the authors are detailing a classical spring-mass-damper, in order to say that some frequencies will be attenuated and a signal recorded. It can be shortened, in my opinion;

especially by comparison with the section 3.1 that would deserve more details. There is a lack of consistency in the mathematical symbols used (multiplication, for example) and citing style.

Fig 4 & 5 are not clear, the double scale of the vertical axis makes the reading difficult. It would be clearer with a log scale or zoom box.

**Response:**

**We apologize for our oversights. These issues have been revised as follows.**

(1) **We have added a new figure (Fig 1.) in the revised manuscript to illustrate the structural relationships between the laser beam, the FB, the cart, the servo motor PVIS, and the reference corner-cube.**

(2) **Unfortunately, we could not find adequate scope for reducing the length of Section 2. However, we added additional details in Subsection 3.1 to better clarify important aspects of our work. Could please help give some suggestions? Thank you!**

(3) **We have addressed inconsistencies in mathematical symbols and reference citation styles throughout.**

(4) **Both Figs. 4 and 5 have been reformatted in the revised manuscript to provide a clearer presentation of results.**

**2, Questions: (specific comments)**

The proposed method is a combination of 2 sub-techniques (PVIS and VECA). But the comparison is global, so it is hard to conclude about the correctness of the sub technique themselves.

What is the sensibility and precision of a PVIS? Can it be compared alone with other techniques?

What if we add prior form of the noise from PVIS into the LSS? Can the VECA be compared with other algorithms taking into account the vibrational signal from the PVIS? l 136. Why add a white noise? Why with this specific variance? l 186.

Why use the genetic algorithm among many others? Gradient? Random walk? Or Machine Learning? l 192.

Why no noise from the fringe signal is considered? l 220.

What are the input and output of the genetic algorithm? What is the variable? Individuals, etc. l 225.

What is exactly the objective function chosen which "is based on the correlation coefficients"? l 226.

In figure 3, in "Is i greater than the maximum number of iterations" is the same i-th time iteration or the i-th iteration of the genetic algorithm? It is confusing. l 236.

The LSS used is Linear or Non-linear ?

**Response:**

**We apologize for our oversights. These issues have been revised as follows.**

**(1) In an effort to address the reviewer's initial comment, we must first note that current practical applications of PVIS are far from meeting the requirements of instrument stability, accuracy, and precision under different observation conditions, as stated in the introduction. In fact, the calibration error of a standard PVIS used in actual measurements is much greater than the allowable error range of a microgravity absolute gravimeter. As such, PVIS alone is not sufficient for addressing the effects of vibrations in the measurements of absolute gravimeters. This issue is addressed by the proposed VECA, which must be employed in conjunction with PVIS. In fact, it cannot be applied separately. The point of the results presented are that the integrated application of both PVIS and VECA yields absolute gravimeter measurement accuracies that are greater than PVIS alone, and do meet the requirements of practical applications. This sufficiently validates the proposed work.**

**(2) We apologize for our failure to clarify why we applied a genetic algorithm for solving the VECA. We have clarified this issue in Subsection 3.1 of the revised manuscript.**

**(3) In the process of the simulations, we verified the anti-interference ability of VECA using the vibration interference models given as equations (3.2) and (3.3). The parameters in (3.2) and (3.3) have been assigned by referencing the actual signal we measured using an actual PVIS system. In addition, white noise has been added to the vibration models.**

**(4) We also apologize for failing to explain how the VECA solution process is initially conducted and clarify the nature of the objective function. We have clarified these issues in Subsection 3.1 of the revised manuscript.**

**(5) We have clarified the confusion regarding the index *i* in the original Fig. 3 by revising this as an index *N* in the revised manuscript, which is now given as Fig. 4.**

**(6) Finally, the LSS employed was linear. This was clarified in the introduction of the revised manuscript.**

**3, Technical corrections/typing errors:**

l 102. nPhi_1 the subscript missing.

l 104, unwanted space before the coma.

l 107. nPhi_2 the subscript missing.

l 119. idem

l 143. k - th

l 147 148. 10 the exponent missing.

l 161. parentheses for cos and sin functions and multiplication sign

l 162. it is equation 2.10 instead of 2.9

l 168. Problems with ))

l 171. g_0 the subscript missing.

l 179. it should d+e instead of d-e?

l 203 204. [ instead of (

l 210. power

l 214. ) instead of ]

**Response:**

We apologize for the many typing errors. These issues have been corrected in the revised manuscript.

---

## Author Comment (AC2) · 15 Feb 2021

Dear Anonymous Referee,

We thank you for your helpful comments and analysis. We have read your suggestions very carefully and have responded to each one below in bold font. All revised portions of the manuscript have been given in red font.

1, (16-17-18) sentence is not clear: the resolution of the tests results shows...the word 'resolution' should be precised as the measuring system resolution. As said, it can be confused with the resolution of the system of equations.

2, (44) (53-54) (238-240) precise the definition of accuracy and precision to avoid misunderstanding.

**Response:**

**Yes, these instances are very confusing. We apologize for our oversight. These two instances were addressed in the revised manuscript.**

3, (66-67) I don't see the relation between the FB –cart motion matching and the time-displacement coordinates measurements. A global schematic of the system could help.

**Response:**

**Yes, the meaning is very unclear. We have added a new figure (Fig 1.) in the revised manuscript to illustrate better the structural relationships between the laser beam, the FB, the cart, the servo motor PVIS, and the reference corner-cube.**

4, (68-71) sentence comprehensible but should be split in two more concise sentences.

**Response:**

**We apologize for this oversight. This has been addressed in the revised manuscript.**

5, (80) applies no closed-loop feedback at any point: need some explanations

**Response:**

**Indeed, this issue is not particularly clear. We have added additional explanation in Subsection 2.1 of the revised manuscript.**

6, (120-121) explain why: Most of the useful signals related to the disturbance error of the absolute gravity measurement in Z(s) can be recovered by the synchronously output vibration signal X(s). in relation with the previous paragraph.

**Response:**

**Yes, this is not entirely clear. We apologize for our oversight. This issue has been clarified in Subsection 2.1 of the revised manuscript.**

7, (217) [6] has to be explicated

**Response:**

**We apologize for this typing error. This issue has been addressed in the revised manuscript.**

8, (219-221) explain the choice of the parameters and the effects in the genetic algorithm.

**Response:**

**We apologize for our failure to clarify why we applied a genetic algorithm for solving the VECA,**

**and to explain how the solution process is initially conducted. We have clarified these issues in Subsection 3.1 of the revised manuscript.**

9, (222-224) again, a short explanation of genetic algorithm could help the comprehension and shows the advantages of the algorithm with respect to the classical least squares method.
**Response:**
**This issue was addressed according to the preceding response.**

10, (269-273) are copy-paste of (13-18), should be reformulated to avoid the exact repetition.
**Response:**
**We appreciate the reviewer's opinion on this issue. However, we must conclude that an exact repetition of text from the main body of the paper in the abstract is not a particularly onerous condition. We do hope that the reviewer will overlook this small issue since it bears no reflection on the work itself.**

12, The following are purely typing corrections:
 (14) (1 Gal = 110-8m/s2) the number 2 should be exponent.
 (42) technology
 (64) servo motor installed on the outside of the drop chamber.
 (66-67) will be are introduced.
 (74) is are connected
 (93) separate the comma from the equation to avoid confusion with prime.
 (103) no space after 'the intrinsic frequency'
**Response:**
**We apologize for the many typing errors. These issues have been corrected in the revised manuscript. However, we find that the issue cited at line (42) in the original manuscript does not represent an error. In addition, the issue cited at line (103) involves a space between "the intrinsic frequency" and its mathematical expression** $\omega_0{}^2 = \frac{K}{M}$**, which may have been garbled in the original manuscript.**

---

## Referee Report (RR1)

REVIEW
Title: Vibration error compensation algorithm in the development of the laser interference absolute gravimeter
Authors: Q. Wu, Y. Teng, X. Wang, Y. Wu, and Y. Zhang
Date: 27 Nov 2020

**1. Remark:**

The modifications done by the authors are easing the reading and improving the quality of the article.
The figures are also clearer.

**2. Specific questions:**

(1). Thank you for this reminder. Maybe it can be recalled at the end of section 2.3 to ease the transition between the two techniques.

(2) Indeed, the authors clarified the genetic algorithm employed for solving the VECA. This is appreciable.

(3) The fact that vibration interference from eq (3.2) and (3.3) are referencing the actual signal measured using an actual PVIS system, can be added to the manuscript for clarity.

(4-5) It seems clearer, with this revised manuscript.

Despite, all the good improvements of the authors, I think that the quality of the manuscript would be really increase by answering the 2 following questions.
(a) Can the PVIS prior (vibration information) be included in the LSS for comparison ?
(b) Can you try to solve the VECA with other techniques ? If yes, make a comparison, if no, you should explain why in the manuscript ?

**3. Typying errors:**
 l. 166 There is still a little inconsistency in the mathematical symbols: * should be replaced by \times, or better 1* can be removed.

---

## Author Response (AR2)

Dear Anonymous Referee,

We thank the referee again for your comments. We have responded to each comment below in bold font. All revised portions of the manuscript have been given in red font.

**1, Remarks**

(3) The fact that vibration interference from eq (3.2) and (3.3) are referencing the actual signal measured using an actual PVIS system, can be added to the manuscript for clarity.

**Response:**

**Unfortunately, the purpose of the referee's comment is not entirely clear. Please note that each of Eqs. (3.2) and (3.3) were employed as a known vibrational interference coupled to the measurement system, which is denoted as $d(t_i)$ in Eq. (2.7). These known interferences were applied for demonstrating the accuracy of the proposed error compensation method because $d(t_i)$ is not generally known in advance. We hope this addresses the referee's comment.**

**2, Specific questions**

(a) Can the PVIS prior (vibration information) be included in the LSS for comparison?

**Response:**

**The displacement of the FB as a function of time was established by Eq. (2.7). Accordingly, the results obtained using the LSS to solve this equation will include not only the quadratic term arising from acceleration due to gravity but also the quadratic term in the vibration interference. However, the PVIS is employed only as a vibration isolation device. Therefore, other algorithms such as the VECA must be employed to reduce the effect of the quadratic term in the vibration interference, and thereby enable the PVIS to be applied simultaneously as both a vibration isolation device and as a sensor detecting the vibration signal of the reference corner-cube.**

(b) Can you try to solve the VECA with other techniques? If yes, make a comparison, if no, you should explain why in the manuscript?

**Response:**

**A discussion of this question has been provided in the revised manuscript.**

**3, Typing errors:**

l. 166 There is still a little inconsistency in the mathematical symbols: * should be replaced by \times, or better 1* can be removed.

**Response:**

**We apologize for these errors. This issue has been corrected in the revised manuscript.**